# OPTIMIZED EARLY-EXIT BASED SPECULATIVE DECODING VIA PIPELINE PARALLELISM

## ABSTRACT

Large language models (LLMs) deliver impressive generation quality, but incur very high inference cost for the auto-regressive decoding manner. Early-exit based speculative decoding (EESD) has emerged to reduce decoding latency. However, in practice, many approaches struggle to achieve an expected acceleration in the draft-then-verify paradigm even with a well-aligned early-exit head and selected exit position. Our analysis reveals that EESD only pays off when the vast majority of draft tokens are accepted by the LLM. Otherwise, the draft cost may overcome the acceleration gain and lead to a negative speedup. To mitigate this, we propose **Pipeline-Parallel Speculative Decoding (PPSD)** that fully pipelines the draft and verification work so that no effort is wasted on failed predictions. It has two key innovations. **Pipeline-Parallel Early-Exit Execution:** We design a fine-grained pipeline allocation and execute system, in which early-exit (draft) computations and remaining-layer (verification) computations overlap with minimal blocking waste. **Verify-while-draft Decoding:** We interleave drafting and verification per token. While the LLM is verifying the current token in its final layers, the early-exit path simultaneously drafts the next token. This high parallel scheme keeps all units busy and validates tokens on-the-fly, analogous to pipelining the speculation and verification stages. Each token is confirmed as soon as it enters the output, ensuring correctness without stalling. All these design choices are supported by both theoretical analysis of pipelined throughput and extensive experiments. Empirical results confirm that PPSD achieves state-of-the-art acceleration in self-speculative LLM inference. On diverse benchmarks, PPSD achieves speedup ratios in the range of $2.01\times \sim 3.81\times$, which gains almost the optimal acceleration at the fixed acceptance rate and exit position, showcasing its advancement in providing efficiency.

## 1 INTRODUCTION

Large Language Models (LLMs) have achieved state-of-the-art performance across a wide range of language tasks, including chatbots (OpenAI, 2024; Yang et al., 2024) and long-term reasoning (Wei et al., 2022; OpenAI, 2024). However, inference with LLMs remains time-consuming and computationally expensive, not only due to their massive parameter sizes but also because of the inherently sequential nature of auto-regressive token generation—each token must be generated based on all preceding tokens. To address this bottleneck, researchers have proposed speculative decoding (SD) (Leviathan et al., 2023; Chen et al., 2023), an effective technique that introduces a smaller draft model to generate a sequence of tokens, which are then verified in parallel by the target model. This draft-then-verify paradigm leverages the efficiency of the smaller model to reduce latency while maintaining the output quality of the target model. Traditional SD typically requires two separately trained models with well-aligned output distributions to realize substantial speedups, which is difficult in practice due to discrepancies between models of different sizes or training regimes.

To overcome this challenge, recent advances in self-speculative decoding (Self-SD) demonstrate that the LLM itself can be reused to both draft and verify tokens (Liu et al., 2024b; Cai et al., 2024; Elhoushi et al., 2024; Xia et al., 2024a). Medusa (Cai et al., 2024) and EAGLE (Li et al., 2024; 2025) generate multiple tokens in a single forward pass and validate them in the subsequent step. Another promising solution, Early-Exit based Speculative Decoding (EESD), re-purposes the first-$E$ layers

of the model to act as the draft model, achieving $1.5\times \sim 2.0\times$ speedup across various tasks (Liu et al., 2024a;b). Apart from avoiding deploying separate models, EESD also allows partial reuse of computation, such as shared activations and KV caches, between the draft and verification stages.

The performance of EESD hinges on several factors: the early-exit position (which affects draft speed), the draft accuracy (i.e., token acceptance rate), and the number of drafted tokens per step (draft length). The expected speedup ratio can be formulated by the aforementioned factors in mathematics. Notably, a trade-off exists that more layers involved in drafting improve the acceptance rate but also increase computational cost (Zarch et al., 2025). Techniques like self-distillation (Zhang et al., 2019; Zhou et al., 2023) and joint optimization of exit positions and draft lengths (Liu et al., 2024c; Sadhukhan et al., 2024; Liu et al., 2024d) attempt to mitigate this trade-off prior to deployment. Nevertheless, in practice, these approaches still struggle to achieve the expected acceleration due to an inappropriate draft length. The acceleration decreases due to the cost of speculation. Contemporary prices exceed gains, and speculative decoding actually causes more latency consumption. If the target model rejects one draft token, all subsequent tokens become invalid, leading to wasted computation and reduced overall speedup.

In this work, we propose Pipeline-Parallel Speculative Decoding (PPSD)—a novel scheme that maximizes acceleration gains while addressing the limitations of draft length sensitivity in EESD. The key innovation lies in two points. Firstly, PPSD executes an early-exit stage based on parallel computing with fully aligned pipelines, which provides not only highly computation resource utility but also immediate draft token verification. Secondly, PPSD implements a verify-while-draft self-speculative decoding paradigm, where each draft token is involved in the next token prediction while being verified through a full model forward in parallel. Benefiting from these designs, PPSD implements early-exit based self-SD with an optimized acceleration gain that mitigates the impact of draft length under a theoretical guarantee. Our main contributions are as follows:

- **Pipeline-parallel early-exit execution:** PPSD leverages a fully-aligned parallel pipeline to execute early-exit and full model forward computations with high hardware utilization while simultaneously verifying tokens, minimizing blocking bubbles.

- **Verify-while-draft decoding:** Instead of waiting to verify an entire draft sequence post-hoc, PPSD enables each draft token to participate in the next token prediction while being verified in parallel by the full LLM immediately. This interleaved execution reduces the risk of accumulating invalid tokens and improves decoding efficiency with eliminated cost.

- **Theoretical and empirical guarantee:** We provide an analysis showing that PPSD achieves better speedup than traditional EESD by mitigating the impact of speculative failure. Experiments across various tasks and LLMs demonstrate that PPSD consistently delivers higher efficiency and reduced latency.

## 2 RELATED WORKS

**Speculative Decoding.** Due to the auto-regressive nature of transformer decoder architectures, LLM inference speed is inherently limited by the sequential token generation process (Xia et al., 2024b). Speculative decoding (Stern et al., 2018; Leviathan et al., 2023; Chen et al., 2023) mitigates this bottleneck by employing a lightweight auxiliary model to draft tokens with low latency, which are then validated in parallel by the target LLM. Subsequent works aim to further improve speedup through enhanced sampling strategies (Miao et al., 2024; Li et al., 2024), soft verification mechanisms (Kim et al., 2024; Sun et al., 2024), and alignment-based learning for draft model extraction. The key acceleration gain lies in its ability to enhance parallelism in each decoding step, significantly accelerating inference without compromising output quality. However, this acceleration relies on the draft model generating tokens from a distribution well-aligned with the target LLM, posing challenges when deploying speculative models across different LLMs.

**Self-Speculative Decoding.** Recent studies (Cai et al., 2024; Fu et al., 2024; Liu et al., 2024b; Li et al., 2024; Xia et al., 2024a) demonstrate the feasibility of speculative decoding using the LLM itself. A representative example is Medusa (Cai et al., 2024), which predicts several subsequent tokens in one forward using trained 'medusa heads'. EAGLE (Li et al., 2024; 2025) further explores the potential of multiple token prediction within a single LLM forward pass. Techniques like early-exit (Liu et al., 2024b; Xia et al., 2024a), layer skip (Elhoushi et al., 2024; Zhang et al., 2024),

and quantization (Zhao et al., 2024; Tiwari et al., 2025) have the potential to support self-SD. Our work leverages early-exit models as drafts to enable self-speculative decoding, achieving accelerated inference without compromising accuracy. The most closely related approach is EESD (Liu et al., 2024b), which employs a fixed exit as the draft model and uses an iterative probabilistic exit mechanism to determine when to exit. In contrast, our proposed pipeline-parallel verify-while-draft mechanism supports early-verify with no extra cost for each draft token, enhancing robust early-exit self-speculation across various LLMs and downstream tasks.

**Pipeline Parallel.** In pipeline parallelism, the model layers are partitioned into sequential segments across multiple computing workers, with micro-batches flowing through these segments to overlap computation and communication, and thereby enable scaling to very large networks (Narayanan et al., 2019). Classic systems exploit this scheme in training, such as GPipe (Huang et al., 2019), PipeDream (Narayanan et al., 2019). More recent work has adapted these ideas to LLM inference (Timor et al., 2024; Blagoev et al., 2025; Hooper et al., 2023). As for SD, PipeInfer (Butler et al., 2024) executes speculative drafting and verification in parallel pipelines, and SPEED (Hooper et al., 2023) runs predicted future-token passes in parallel with current-token computation. Other works like DSI (Timor et al., 2025) and PEARL (Liu et al., 2025) use multiple verifier, that executed in parallel, to speedup SD. However, they often introduce huge pipeline bubbles, and thus fall short in application. Early-exit frameworks also leverage pipeline schedules. EE-LLM (Chen et al., 2024) uses 3D-parallelism to train and serve large early-exit LLMs with negligible overhead. Together, these results show that pipelined execution can orchestrate multi-stage inference (e.g. overlapping early-exit or draft/verify phases), motivating our proposed PPSD with a verify-while-draft scheme.

## 3 PRELIMINARIES

### 3.1 EARLY-EXIT BASED SPECULATIVE DECODING

Speculative decoding (Leviathan et al., 2023; Chen et al., 2023) provides efficient LLM decoding with a small draft model $\mathcal{M}_d$ generating several draft tokens and the target model $\mathcal{M}_t$ validating in parallel. Given fixed draft length $\gamma$ with input sequence $[\boldsymbol{x}_1, \boldsymbol{x}_2, \cdots, \boldsymbol{x}_n]$ and pre-generated token sequence $[\boldsymbol{y}_1, \boldsymbol{y}_2, \cdots, \boldsymbol{y}_j]$, the draft model generates the next token by

$$\boldsymbol{d}_h \sim p_{j+h} \leftarrow \mathcal{M}_d(\boldsymbol{d} \mid \boldsymbol{x}_{\leq n}, \ \boldsymbol{y}_{\leq j}, \ \boldsymbol{d}_{<h}), \tag{1}$$

where $\boldsymbol{d}_h$ represent the $h$-th draft token, $h \leq \gamma$, and $p_h$ refers to the probability distribution, from which $\boldsymbol{d}_h$ is sampled. Then the target model $\mathcal{M}_t$ validates all the $\gamma$ draft tokens in parallel based on the following strategy:

$$\mathbb{P}(\boldsymbol{y}_{j+h} \leftarrow \boldsymbol{d}_h) = \mathbb{P}\left(r \leq \min\left(1, \frac{q_{j+h}(\boldsymbol{d}_h \mid \boldsymbol{x}_{\leq n}, \ \boldsymbol{y}_{\leq j}, \ \boldsymbol{d}_{<h})}{p_{j+h}(\boldsymbol{d}_h \mid \boldsymbol{x}_{\leq n}, \ \boldsymbol{y}_{\leq j}, \ \boldsymbol{d}_{<h})}\right)\right), \tag{2}$$

where $r \sim U[0,1]$ is randomly sampled from a uniform distribution, $q_{j+h} \leftarrow \mathcal{M}_t(\boldsymbol{y} \mid \boldsymbol{x}_{\leq n}, \ \boldsymbol{y}_{<j}, \boldsymbol{d}_h)$ denotes the probability distribution of $\boldsymbol{y}_j$ computed by $\mathcal{M}_t$. If exactly $i$ draft tokens are accepted (and token $i+1$ is the first rejected token, for $0 \leq i < d$), then those $i$ tokens become part of the output along with one additional token from the target model, while the left $d-i$ drafts are discarded. If all $d$ draft tokens are accepted (probability $\alpha^d$), then $d$ tokens are added plus one final token, for a total of $d+1$ output tokens. Given input sequence length $s$, let $\mathcal{T}_t(s)$ and $\mathcal{T}_d(s)$ denote the time for the target and draft model to forward once, respectively. The acceleration gain from SD can be formulated as

$$\rho := \frac{\mathcal{T}_{\boldsymbol{T}}(1)}{\gamma \mathcal{T}_{\boldsymbol{D}}(1) + \mathcal{T}_{\boldsymbol{T}}(\gamma)} \cdot (\mathbb{E}(\gamma) + 1) \tag{3}$$

Here $\mathbb{E}(\gamma)$ refers to the expected accept token length with $\gamma$ draft tokens. Let $\alpha$ represent the acceptance rate of drafted tokens from gold prefix sequences, that is, the case of draft length $\gamma = 1$. Suppose that the target model accepts the draft token with fixed probability $\alpha$, we can derive the expected accepted draft sequence length as

$$\mathbb{E}(\gamma) := \sum_{h=0}^{\gamma-1} h\alpha^h(1-\alpha) + \gamma\alpha^\gamma = \frac{\alpha(1-\alpha^\gamma)}{1-\alpha}.$$

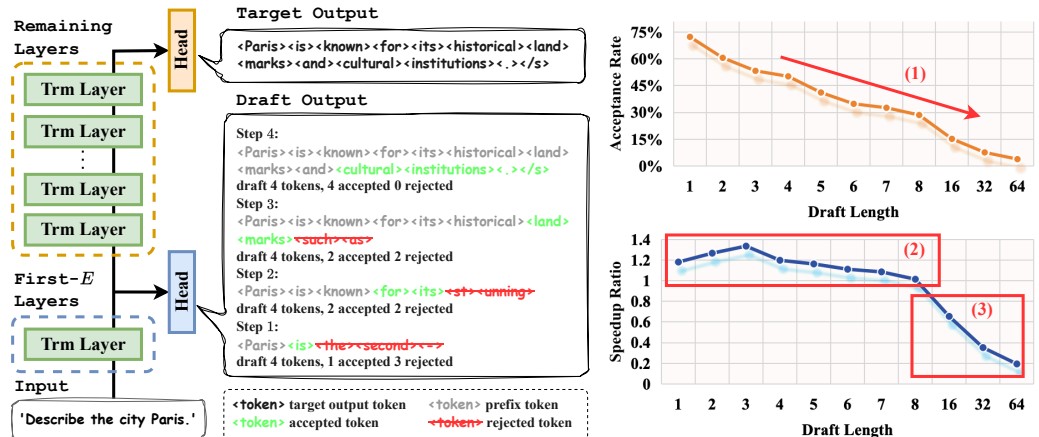

Figure 1: The left side of the figure shows a generation example from EESD using a draft-then-verify paradigm, where the draft length is fixed at 4. It can be observed that the majority of draft tokens at each step are invalid (i.e., rejected by the target model), resulting in additional draft overhead and increased decoding latency. The right side of the figure presents the acceptance rate and speedup ratio under different draft lengths. As shown in (1), the acceptance rate noticeably decreases with longer drafts. In (2), the speedup ratio initially increases, indicating a positive acceleration gain. However, as the draft length continues to grow, the declining acceptance rate introduces greater draft overhead, ultimately offsetting the speedup ratio.

Since the computation is not the bottleneck in the decoding phase, we assume that $\mathcal{T}_T(\gamma) = \mathcal{T}_T(1)$.

As for early-exit supported self-SD (Liu et al., 2024b;a), the draft token is predicted with hidden states from the first $E$ layers of the target LLM. Notably, the verification procedure reuses cached activations from the draft phase: only the remaining $N - E$ layers must be computed. Thus, the time to draft $\gamma$ tokens is proportional to the time to verify them by forwarding the remaining $N - E$ layers (in a single batch, reusing KV caches). The early-exit head is often set to an aligned network with the target output head, such as an RMSNorm layer subsequent by a linear layer. With only the forward pass time considered, the corresponding rate $\rho$ is as following

$$\rho = \frac{(1 - \alpha^{\gamma+1})N}{(1 - \alpha)(\gamma E + N)}, \tag{4}$$

where $E \leq N$, and $N$ refers to the total layer number of the target model.

## 3.2 MOTIVATING OBSERVATIONS

It is evident in equation 4 that the acceleration gain from early-exit decoding is influenced by several key factors: the prefixed exit position $E$, the draft token acceptance rate $\alpha$, and the draft length $\gamma$. To maximize inference speedup, prior works have widely investigated how to extract an early-exit head that generates tokens closely aligned with the target LLM output distribution (Liu et al., 2024b; Elhoushi et al., 2024), thereby improving acceptance rate $\alpha$. However, both $\alpha$ and $E$ are typically fixed settings through careful tuning and alignment learning, and their optimal values may vary across models and downstream tasks (Zarch et al., 2025). In practical deployments, determining an optimal $\gamma$ is equally critical within the draft-then-verify paradigm, yet this dimension remains under-explored in existing works.

From equation 4, the draft length $\gamma$ influences the acceleration gain through the term $\frac{1-\alpha^{\gamma+1}}{\gamma}$. Although the numerator asymptotically approaches 1 as $\gamma$ increases, the denominator $\gamma$ grows linearly, causing the overall value to decrease beyond a certain point. This behavior also aligns with practical observations. As illustrated in Figure 1, a case study in EESD under the draft-then-verify paradigm shows that when drafting 4 tokens per step, many tokens are rejected during verification, rendering the subsequent drafts invalid. This paradigm lacks a mechanism for detecting drafting failures,

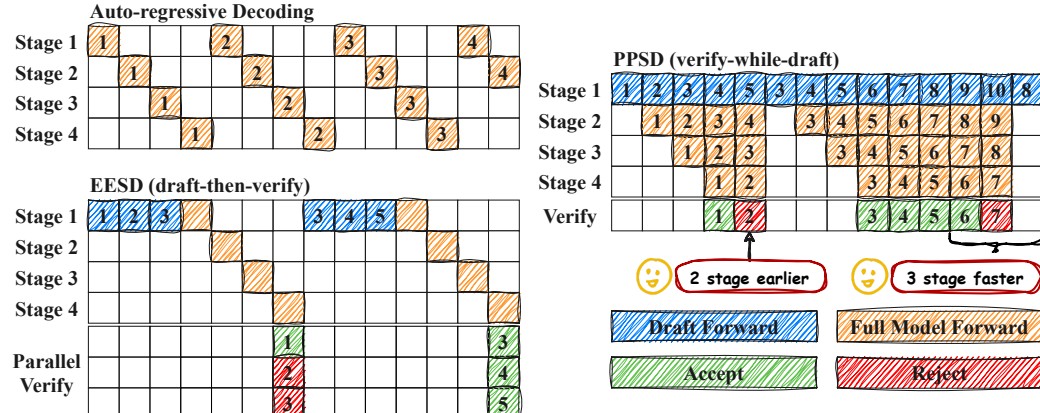

Figure 2: This figure compares the LLM inference pipelines of auto-regressive decoding, EESD, and our proposed PPSD. Both auto-regressive decoding and EESD, which adopt the draft-then-verify paradigm, suffer from significant computational idling. Moreover, the draft-then-verify approach lacks a mechanism to identify tokens likely to be rejected, resulting in numerous wasted draft attempts and additional latency. In contrast, our proposed PPSD achieves acceleration in two ways. It adopts a verify-while-draft scheme, which allows each draft token to be verified without additional cost. Acceleration is further achieved through pipeline-parallel execution. As show in the case, PPSD rejects the token '2' two stages earlier and completes three stages faster than EESD.

leading to substantial overhead. We formulate the overall acceptance rate as bellow:

$$\alpha_{\text{all}}(\gamma) = \frac{\mathbb{E}(\gamma)}{\gamma} \leq \frac{\alpha(1 - \alpha^{\gamma})}{(1 - \alpha)\gamma}. \tag{5}$$

According to equation 5, for fixed acceptance rate $\alpha$, the overall acceptance rate $\alpha_{\text{all}}$ decreases with larger $\gamma$, which is consistent with empirical results in Figure 1, highlighting how cumulative distributional error degrades draft efficiency. From a decoding speed perspective, longer drafts often provide diminishing or even negative returns due to frequent rejections, increasing overall cost. Motivated by this, our work focuses on mitigating the adverse effects of $\gamma$.

## 4 METHODS

In section 3, we have indicated that the acceleration gain degrades due to the accumulated invalid speculation. In this section, we introduce **PPSD**, a **P**ipeline **P**arallelism based infrastructure for early-exit supported self-**S**peculative **D**ecoding that provides optimized acceleration gain with mitigated draft-then-verify cost.

### 4.1 PIPELINE PARALLEL EARLY-EXIT EXECUTION

We first introduce the stage-level model deployment with pipeline parallelism. The first $E$ layers with the early-exit head having aligned output performance serve as the draft model, which determines the granularity of pipeline parallel computing allocation. The whole model with $N$ layers is divided into $\lceil N/E \rceil$ stages. The last stage includes not only the last few transformer layers but also the language model head sampling the target output. Each stage is executed by a pipeline worker with independent computation resources. It is worth noting that if $N$ can not be divided by $E$, the remaining layers are still encapsulated as an independent stage to support pipeline parallelism without extra bubbles from unbalanced computation distribution. Besides, for a deep exit with large $E$, we can introduce a smaller $S$, where $S < E$ and $S$ divides $N$, so as to minimize blocking bubbles in the pipeline. Such deployment design matches the LLM application setting when serving for a large batch size or just deploying a large model with huge weights that can not fit in a single GPU memory, where model layers are evenly allocated in different computation units.

During the token generation phase, the overall pipeline parallel system executes as shown in Figure 2. The first worker proposes draft tokens simultaneously and transmits the activations to the next

worker. The target model forward pass is executed sequentially, and the last worker samples the target model output at the end. After warm-up, all the workers can execute computation in parallel, thereby achieving high GPU usage and parallelism.

## 4.2 VERIFY-WHILE-DRAFT DECODING

As shown in Figure 2, both auto-regressive decoding and previous EESD methods operate in a sequential, draft-then-verify manner, resulting in underutilized GPU resources and an inability to detect invalid drafts and their associated overheads. The weakness is clearly illustrated with both the theoretical analysis and empirical results. In PPSD, we implement a verify-while-draft paradigm to support self-SD with fast drafting and parallel verification. Benefiting from the well-aligned computation burden in each worker, the whole system can execute with minimal bubbles. The draft worker keeps sampling draft tokens while the full model verifies them in parallel. Each draft token can be verified within $\lceil N/E \rceil$ stages, which takes the same amount of time as the target model decoding one token through a full forward pass. Therefore, PPSD does not introduce extra drafting or verification costs for re-computation or waiting. The acceleration gain from PPSD is given by

$$
\begin{aligned}
\rho^\star :=& \frac{\text{\# tokens per sec (PPSD)}}{\text{\# of tokens per sec (auto-regressive)}} \\
=& \frac{1/\mathcal{T}_D}{1/(\alpha \frac{E}{N}\mathcal{T}_D + (1-\alpha)\lceil \frac{N}{E}\rceil \frac{E}{N}\mathcal{T}_D)} \\
=& \frac{N}{\alpha E + (1-\alpha)\lceil \frac{N}{E}\rceil E}.
\end{aligned}
\tag{6}
$$

According to equation 6, when the draft early-exit head performs perfect speculative sampling with $\alpha = 1$, PPSD can achieve $\rho = N/E$ speedup ratio. Even in the worst case with acceptance rate $\alpha = 0$, the acceleration gain reduces to $\rho = 1$, which is aligned with auto-regressive decoding speed with no extra draft cost. Compared to the vanilla version of EESD, $\rho^\star$ exceeds $\rho$ by

$$
\lambda = \frac{\rho^\star}{\rho} = \frac{(1-\alpha)(\gamma E + N)}{\left[\alpha E + (1-\alpha)\lceil \frac{N}{E}\rceil E\right](1-\alpha^{\gamma+1})}.
\tag{7}
$$

It is evident that $\lambda > 1$, showcasing an improved speedup performance. To be specific, PPSD provides, firstly, the early verification that avoids over-drafting. As seen in Figure 2, the verification of the 2-nd token is 2-stage earlier than that of EESD with draft-then-verify routine. Secondly, the high parallelism means verification does not need to recompute. When $\alpha = 1$, we have $\lambda = \gamma + N/E$, which is the extra acceleration gain from pipeline parallelism.

## 5 EXPERIMENTS

In this section, we demonstrate the effectiveness of the proposed PPSD, showcasing its ability to mitigate draft cost and increase optimal speedup for EESD, which matches our theoretical analysis.

### 5.1 EXPERIMENTAL SETUP

**Implementation Details.** We evaluate PPSD on the Vicuna v1.5 series with 7B and 13B model sizes (Zheng et al., 2023), LLaMA-2 13B model, and the extremely parameterized LLaMA-2 70B model (Touvron et al., 2023). To make early-exit suitable for self-SD, we train early-exit heads by self-distillation to align the draft and target output distribution. The training and subsequent evaluation for the Vicuna 7B model is completed by 8 A100 40GB GPUs, and for Vicuna 13B, LLaMA-2 13B and 70B, we use 8 H100 80GB GPUs. Note that during evaluation, the exact amount of GPU resources depends on model size and number of workers used corresponding to PPSD granularity, and will be specified later. The evaluation benchmarks across various tasks include XSum (Narayan et al., 2018) for document summation, GSM8K for mathematical reasoning (Cobbe et al., 2021), and HumanEval (Chen et al., 2021) for code generation ability. The maximum output token length in all the benchmarks is set to 512. To better evaluate the acceleration from speculative decoding methods, we execute inference with a fixed batch size of 1.

Table 1: Evaluation on XSum, Gsm8k and HumanEval with different methods. **AR** refers to the overall acceptance rate. **GS** signifies the token generation speed in number of tokens per second. **SR** means the wall-time speedup ratio compared with auto-regressive decoding. **Avg** denotes the average speedup ratio in the three benchmarks. Here 'Auto' denotes the auto-regressive decoding. EESD with superscribe † and ‡ refer to EESD with fixed draft length 5 and 10, respectively, while superscribe 4 and 8 distinguish that EESD utilizes first $1/4$ or $1/8$ model layers as draft model.

| Model | Method | XSum | | | Gsm8k | | | HumanEval | | | Avg |
|---|---|---|---|---|---|---|---|---|---|---|---|
| | | AR | GS | SR | AR | GS | SR | AR | GS | SR | |
| V 7B | Auto | - | 29.32 | $1.00\times$ | - | 25.07 | $1.00\times$ | - | 30.66 | $1.00\times$ | $1.00\times$ |
| | EESD$^{4,\dagger}$ | 36.33% | 26.89 | $0.92\times$ | 46.59% | 28.17 | $1.12\times$ | 52.55% | 36.80 | $1.20\times$ | $1.08\times$ |
| | EESD$^{4,\ddagger}$ | 21.71% | 21.31 | $0.73\times$ | 29.37% | 23.43 | $0.93\times$ | 35.14% | 32.29 | $1.05\times$ | $0.90\times$ |
| | PPSD$^4$ | 67.38% | 53.53 | $1.83\times$ | 71.64% | 48.80 | $1.95\times$ | 80.56% | 70.32 | $2.29\times$ | $2.02\times$ |
| | PPSD$^8$ | 56.75% | 58.05 | $\mathbf{1.98\times}$ | 59.92% | 52.39 | $\mathbf{2.09\times}$ | 71.84% | 78.80 | $\mathbf{2.57\times}$ | $\mathbf{2.21\times}$ |
| V 13B | Auto | - | 29.62 | $1.00\times$ | - | 25.83 | $1.00\times$ | - | 28.98 | $1.00\times$ | $1.00\times$ |
| | EESD$^{4,\dagger}$ | 31.16% | 29.50 | $1.00\times$ | 35.00% | 26.75 | $1.04\times$ | 54.76% | 45.86 | $1.58\times$ | $1.21\times$ |
| | EESD$^{4,\ddagger}$ | 18.73% | 19.78 | $0.67\times$ | 20.25% | 18.25 | $0.71\times$ | 37.14% | 34.56 | $1.19\times$ | $0.86\times$ |
| | PPSD$^4$ | 68.32% | 57.77 | $1.95\times$ | 70.87% | 53.38 | $2.07\times$ | 81.28% | 74.69 | $2.58\times$ | $2.20\times$ |
| | PPSD$^8$ | 63.19% | 66.65 | $\mathbf{2.25\times}$ | 63.56% | 57.60 | $\mathbf{2.23\times}$ | 75.82% | 81.72 | $\mathbf{2.82\times}$ | $\mathbf{2.43\times}$ |
| L 13B | Auto | - | 17.54 | $1.00\times$ | - | 13.03 | $1.00\times$ | - | 15.59 | $1.00\times$ | $1.00\times$ |
| | EESD$^{8,\dagger}$ | 49.01% | 22.92 | $1.30\times$ | 20.23% | 18.01 | $1.38\times$ | 59.59% | 27.19 | $1.74\times$ | $1.47\times$ |
| | PPSD$^8$ | 74.49% | 48.69 | $\mathbf{2.77\times}$ | 69.97% | 34.58 | $\mathbf{2.66\times}$ | 85.33% | 59.50 | $\mathbf{3.81\times}$ | $\mathbf{3.08\times}$ |
| L 70B | Auto | - | 9.66 | $1.00\times$ | - | 8.05 | $1.00\times$ | - | 8.55 | $1.00\times$ | $1.00\times$ |
| | EESD$^{8,\dagger}$ | 26.75% | 7.62 | $0.79\times$ | 31.83% | 8.08 | $1.00\times$ | 47.05% | 9.03 | $1.06\times$ | $0.95\times$ |
| | PPSD$^8$ | 57.41% | 19.61 | $\mathbf{2.03\times}$ | 57.44% | 15.94 | $\mathbf{1.98\times}$ | 58.48% | 17.37 | $\mathbf{2.03\times}$ | $\mathbf{2.01\times}$ |

**Baselines.** In our main experiments, we compare PPSD to auto-regressive decoding and vanilla EESD with fixed draft lengths of 5 and 10. We also conduct a comparison with EESD with dynamic draft control strategies like Thompson sampling (Liu et al., 2024b) to show the superiority of PPSD through the verify-while-draft paradigm.

**Evaluation Metrics.** We report several widely-used metrics for PPSD evaluation: the overall acceptance rate $\alpha_{\text{all}}$ (Chen et al., 2023; Leviathan et al., 2023), the speed of token generation in decoding phase (in tokens/s), and the exact wall-time acceleration gain $\rho$ compared with auto-regressive decoding. The accuracy of PPSD supported LLM inference is aligned with the auto-regressive one with theoretical guarantees and is not reported here for simplicity.

## 5.2 MAIN RESULTS

To demonstrate the effectiveness of PPSD in mitigating draft cost, we implement it against auto-regressive decoding and vanilla EESD with different draft lengths (5 and 10) on XSum, Gsm8k, and HumanEval benchmarks based on different LLMs. Table 1 reports the comparison results. As shown in Table 1, PPSD achieves optimized performance robustly on all three benchmarks with recovered overall acceptance rate and near-optimal speedup ratio in the range of $2.01\times \sim 3.81\times$. Vanilla EESD with inappropriate draft lengths obtains negative acceleration gains due to invalid speculation and the subsequent draft model forward time consumption. For instance, EESD with $\gamma = 5$ has $0.92\times$ speedup ratio and $0.73\times$ when validated after drafting 10 tokens, which are even worse than the auto-regressive case, showcasing a great draft cost for invalid speculation. However, PPSD, with the same $1/4$ model layers to execute speculation, performs effectively and optimizes EESD robustly with $2.21\times$ speedup in average using the Vicuna 7B model and $2.43\times$ based on the Vicuna 13B model. It is worth noting that the overall acceptance rates of EESD cases differ from PPSD with the same exit settings due to accumulated invalid draft tokens. The exact speedup ratio varies under different benchmarks due to the various acceptance rates in different tasks. Early-exit models share a higher acceptance rate under the code generation task, i.e., HumanEval, which could be due to simpler and high-frequency tokens being accepted in coding.

Table 2: The acceleration effectiveness comparison of the proposed PPSD methods with other baselines on GSM8K and XSum based on Vicuna 13B and LLaMA-2 13B. EESD w. TS refers to EESD with the Thompson sampling control mechanism.

| Model | Method | XSum | | Gsm8k | |
|---|---|---|---|---|---|
| | | GS | SR | GS | SR |
| V 13B | Medusa | 35.48 | 1.21× | 38.36 | 1.53× |
| | EESD w. TS | 37.24 | 1.27× | 39.87 | 1.59× |
| | PPSD$^4$ | 57.77 | 1.95× | 53.38 | 2.07× |
| | PPSD$^8$ | 66.65 | **2.25×** | 57.60 | **2.23×** |
| L 13B | Medusa | 26.83 | 1.53× | 27.59 | 1.77× |
| | EESD w. TS | 33.67 | 1.92× | 31.80 | 2.04× |
| | PPSD$^8$ | 48.69 | **2.77×** | 59.50 | **3.81×** |

We also conduct a comparison with other self-SD methods, including Medusa (Cai et al., 2024), and EESD supported by a dynamic stopping mechanism, i.e., Thompson Sampling optimized control mechanism (Liu et al., 2024b), to demonstrate the advancement of our proposed PPSD method. We implement inference on Xsum and Gsm8k benchmarks based on the Vicuna 13B model and LLaMA-2 13B model. To ensure fairness, we utilize EESD for both other methods and our proposal with the same generation configurations. Results in Table 2 show that our proposed PPSD achieves state-of-the-art accelerations with $1.95\times \sim 3.81\times$ average speedup ratio. Thompson Sampling provides an optimized control mechanism to decide when to stop and recover the speedup ratio. However, our method directly addresses the bottleneck caused by the draft length and thus obtains an optimal speedup ratio in practice.

## 5.3 Ablation Study

To elucidate the impact of different components within our approach, we conduct a series of ablation studies. PPSD obtains an optimal upper bound of acceleration gain, which comes from two parts: the pipeline parallelism with high computation reuse, and the cost-free verify-while-draft paradigm. We first evaluate the impact of the pipeline parallelism.

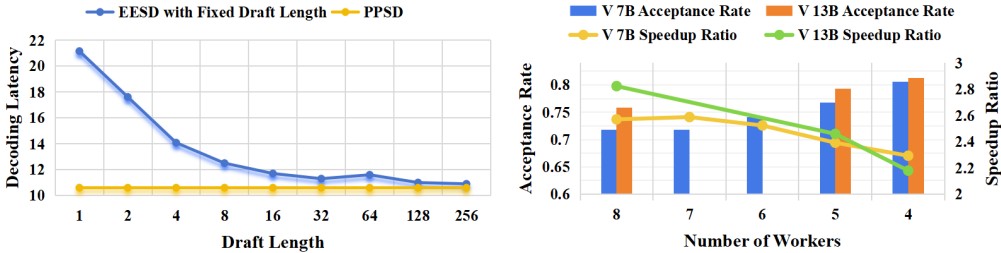

Figure 3: Ablation study on pipeline parallel early-exit execution without verification.

Figure 4: Ablation study on parallel granularity. 'V' refers to the Vicuna series models.

**Impact of pipeline parallelism.** We evaluate the impact of pipeline parallelism using an inference task that generates 512 tokens based on the Vicuna 7B model. In this experiment, the vanilla EESD method is applied with varying draft lengths $\gamma$, ranging from 1 to 256. Both the vanilla EESD and PPSD utilize the first 16 layers of the model—half of the full 32-layer architecture—allowing PPSD to execute forward passes in parallel across 2 workers. The batch size is fixed at 1. To isolate the acceleration benefits of pipeline parallelism, we do not perform any verification in this evaluation. The results are summarized in Figure 3. As expected, longer draft lengths result in lower overall decoding latency for EESD, since more tokens can be speculated and generated in fewer steps. With a batch size of 1, computation remains lightweight even when up to 256 tokens are processed in parallel. Consequently, the decoding latency is primarily determined by the propagation of the early-exit draft generation time and the full model forward pass time, of which the full model pass decreases as the draft length increases. PPSD achieves the shortest execution time, demonstrating

the additional speedup gained through pipeline parallelism. This highlights the effectiveness of overlapping computation across multiple workers during the speculative decoding process.

**Impact of verify-while-draft.** Thanks to the highly aligned pipelines, PPSD brings a verify-while-draft paradigm that allows each draft token to be verified in the continued full model forward pass and also be involved in predicting the next token in parallel, thus obtaining a cost-free verification. The effect of verify-while-draft can be seen in Figure 1, where the overall acceptance rate of PPSD is aligned with the case of $\gamma = 1$ in EESD. As the draft length grows, EESD, performing a draft-then-verify paradigm, gets a decreased overall acceptance rate, which finally leads to a degraded speedup ratio with high draft cost.

**Impact of parallel granularity.** Guaranteed by equation 6, the acceleration gain of PPSD is only influenced by the acceptance rate $\alpha$ and exit position selection $E$, which is a widely discussed trade-off in self-SD (Liu et al., 2024b; Xia et al., 2024b). In the considered pipeline parallel case, the trade-off can be seen as the parallel granularity that determines how much computation workload is evenly allocated to each pipeline worker. We conduct evaluation with different granularity settings based on Vicuna 7B and 13B models. The results are reported in Figure 4.

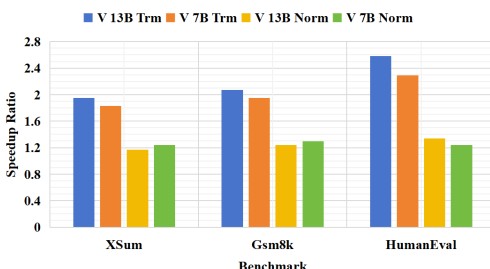

Figure 5: The comparison of norm head and transformer-based head on different benchmarks with the first-$1/4$ layers as draft model.

Table 3: Evaluation results on executing PPSD from different positions with norm head based on Vicuna series models. $E$ refers to the exit position.

| Model | $E$ | XSum | | Gsm8k | | HumanEval | |
|---|---|---|---|---|---|---|---|
| | | AR | SR | AR | SR | AR | SR |
| V 7B | 8 | 32.34% | 1.30× | 32.26% | 1.30× | 22.40% | 1.02× |
| | 16 | 38.48% | 1.18× | 54.54% | 1.35× | 64.03% | 1.44× |
| | 24 | 32.95% | 1.12× | 85.64% | 1.16× | 87.44% | 1.18× |
| V 13B | 10 | 36.67% | 1.17× | 42.15% | 1.24× | 51.81% | 1.34× |
| | 20 | 62.70% | 1.37× | 68.44% | 1.48× | 74.54% | 1.56× |
| | 30 | 84.00% | 1.08× | 85.91% | 1.15× | 89.30% | 1.19× |

**Impact of exit head.** In previous experiments, all the early-exit heads consisted of a single transformer layer followed by a norm head, with the same structure as that of the target model. The optimal acceleration gain of our proposed PPSD relies on a perfectly aligned pipeline scheme, with well-balanced workloads allocated across each worker. An additional head whose parameter size matches that of a transformer layer introduces imbalance in the pipeline. Therefore, we evaluate the effect of using exit heads with different structural configurations. We employ a norm head to assess the performance of a well-aligned pipeline parallelism setup. The results are summarized in Figure 5 and Table 3. When exiting at the same position, norm heads exhibit lower acceptance rates compared to transformer-based heads due to the less parameterized output network. Nonetheless, a notable speedup can still be observed when using norm heads during inference. For instance, when exiting after first $E = 8$ layers of Vicuna 7B and using a norm head for token drafting, the overall acceptance rate is $\alpha = 32.26\%$ in Gsm8k benchmark and a $1.30\times$ speedup is obtained, while the theoretical acceleration gain in equation 6 is $1.31\times$. The results show the ability of PPSD to attain the expected acceleration with introducing additional drafting overhead.

## 6 CONCLUSION

In this manuscript, we presented Pipeline-Parallel Speculative Decoding (PPSD), a novel approach to accelerate auto-regressive inference in LLMs by addressing the inefficiency impact from invalid drafting of EESD. By introducing a pipeline-parallel early-exit execution scheme and a verify-while-draft decoding paradigm, PPSD eliminates the computational cost of failed speculative drafts and improves hardware utilization, all of which bring about optimal acceleration gains for efficient LLM decoding. Theoretically, PPSD optimizes the acceleration gain from mitigated draft length; empirically, it achieves state-of-the-art speedups ranging from $2.01\times$ to $3.81\times$ across diverse tasks and LLM architectures. Our findings demonstrate that PPSD provides a practical and scalable path toward faster LLM inference without compromising output quality.

## 7 Ethic Statement

This work adheres to the ICLR Code of Ethics. In this study, no human subjects or animal experimentation was involved. All datasets used were sourced in the compliance with relevant usage guidelines, ensuring no violation of privacy. No personal identifiable information was used, and no experiments were conducted that could raise privacy or security concerns. We are committed to maintain transparency and integrity throughout the research process.

## 8 Reproducibility statement

We have made every effort to ensure that the results in this paper are reproducible. All code and datasets have been made publicly available in an anonymous repository, which will be specific in the supplementary materials, to facilitate replication and verification. The experimental setup, including training details, model configurations, and inference settings, is described in the paper.

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

## A   LLM Usage

LLMs were used only to aid in the writing and polishing of the manuscript.

## B   Discussion

Our investigation of Pipeline-Parallel Self-Speculative Decoding (PPSD) demonstrates that fully pipelining draft and verification stages can substantially mitigate the cost of invalid speculative tokens, yielding consistent $2.01\times \sim 3.81\times$ speedups across diverse LLMs and tasks. However, several practical considerations and limitations warrant further discussion.

### B.1   Limitations

**Infrastructure Requirements.** PPSD relies on a fine-grained pipeline-parallel deployment across multiple GPUs (or pipeline workers), which may not be available in all inference-serving environments, e.g. resource limited scenarios. Small-scale or single-GPU deployments cannot fully exploit the parallelism advantages and may see only marginal gains.

**Pipeline Granularity Trade-offs.** Determining the optimal number of layers per pipeline stage (i.e., the exit depth $E$ and the number of workers) requires careful tuning: too coarse-grained a pipeline limits overlap opportunities, while too fine-grained a pipeline can introduce communication overhead and imbalance.

**Bounded by Acceptance Rate.** Although PPSD decouples draft length $\gamma$ from throughput degradation, poor alignment (low $\alpha$) still reduces effective speedup (see in 6), since verification must proceed for nearly every draft token. Improvements in early-exit head accuracy (e.g., via stronger distillation in D.1) remain critical.

**Model Architecture Constraints.** Our design targets decoder-only transformer architectures with homogeneous layer structures. Extending PPSD to encoder–decoder models or heterogeneous architectures may require nontrivial modifications to the pipeline schedule and exit-head design.

### B.2   Applications

PPSD is particularly well-suited for high-throughput LLM services —such as real-time chatbots, summarization APIs, and code-generation endpoints—where latency and cost per token are paramount. By seamlessly integrating into existing decoding loops without altering the backbone weights or requiring separate draft models, PPSD can be adopted in production clusters to reduce GPU-hours and energy consumption. Moreover, PPSD's verify-while-draft paradigm could be combined with complementary techniques—such as quantization, multiple token prediction, or dynamic token-level branching—to further compress inference time.

### B.3   Future Direction.

Looking ahead, we plan to explore adaptive pipeline reconfiguration: dynamically altering $E$ and the number of pipeline stages based on runtime acceptance statistics, input complexity, or hardware availability. Another promising avenue is multi-exit speculative decoding, where multiple early-exit heads at different depths collaborate to adaptively adjust the draft length on a per-token basis. When integrated with PPSD, this approach can further reduce the number of discarded draft tokens by dynamically selecting the early-exit head with the highest expected acceptance rate at each step. Finally, integrating soft-verification mechanisms—where partial token probabilities guide speculations—may further improve the acceptance rate $\alpha$ and unlock additional speedups without sacrificing output quality.

## C   Execution of PPSD

The **Pipeline Based Self-Verification Mechanism** in 1 implements our distributed inference framework using a multi-stage pipeline architecture for token generation and verification. Beginning with

input sequence $x_T$, the algorithm performs iterative token generation through coordinated computation across $N$ GPU devices. The core workflow comprises three phases:

1. **Master control initialization**: $GPU_0$ initiates generation and performs first-stage inference, establishing the pipeline via cross-device activation passing.

2. **Intermediate processing**: Nodes $GPU_1$ to $GPU_{N-1}$ progressively process features, generating verification tokens $t_{i+1}^j$ upon detecting early-exit conditions.

3. **Final verification**: $GPU_{N-1}$ serves as the terminal verification layer, transmitting either:

$$\text{TokenType} = \begin{cases} \text{standard tokens} & \text{if NOT} \quad \text{HAS\_EARLY\_EXITED} \\ \text{check tokens} & \text{otherwise} \end{cases}$$

# D    TRAINING DETAILS

## D.1    DESIGN OF SELF-DISTILLATION LOSS FUNCTION

To maximize speedup, it is crucial to select an early-exit point that occurs sufficiently early while preserving an acceptable token acceptance rate. An early-exit that is too shallow may produce low-quality drafts with low acceptance, while one that is too deep undermines the potential speed gain. To achieve this, we adopt a self-distillation mechanism to train the early-exit heads. The training objective is defined as follows:

$$L = \lambda \sum q(\boldsymbol{x}) \log \left( \frac{q(\boldsymbol{x})}{p(\boldsymbol{x})} \right) + H(p(\boldsymbol{x}), \boldsymbol{y}),$$

where the first term is the Kullback-Leibler (KL) divergence between the teacher model's output $q(\boldsymbol{x})$ and the student (early-exit head) output $p(\boldsymbol{x})$, encouraging the early-exit head to mimic the final model's predictive distribution. The second term, $H(p(\boldsymbol{x}), \boldsymbol{y})$, is the standard cross-entropy loss between the student prediction and the ground-truth label $\boldsymbol{y}$. The hyperparameter $\lambda$ controls the trade-off between imitating the teacher and fitting the true labels.

In practice, this joint optimization helps the early-exit head learn both generalizable representations (from the teacher) and task-specific correctness (from labels), ensuring it captures the semantics of the final output without requiring full forward computation. This makes the early-exit head more reliable and accurate, leading to higher-quality drafts and improved inference efficiency under speculative decoding. During training, the teacher logits $q(\boldsymbol{x})$ are obtained by forward-passing the same inputs through the final output head of the backbone model. This process introduces no architectural changes and is compatible with any decoder-only LLM backbone.

## D.2    EXECUTION DETAILS

We conduct all experiments using the PyTorch deep learning framework on NVIDIA A100-40G and H100-80G GPUs. Specifically, we utilize A100-40G GPUs to train early-exit heads for Vicuna-7B, while H100-80G GPUs are used for training with Vicuna-13B, LLaMA-2-13B, and LLaMA-2-70B models.

For the norm head variant, we set the learning rate to 2e-5 and use a batch size of 16. For the transformer-based head, we adopt a learning rate of 5e-4 and a batch size of 2. A comprehensive summary of all hyperparameter configurations is provided in 4.

# E    EVALUATION DETAILS

## E.1    MODEL CONFIGURATIONS

We evaluate our method on a set of representative models, including LLaMA-2 and Vicuna-v1.5. Detailed model configurations are summarized in 5.

---

**Algorithm 1** Pipeline Based Self-Verification Mechanism

---

**Input:** sequence $x_T$

1: $s_0 \leftarrow \{x_T\}$
2: **for** $i \leftarrow 0, 1, 2, \ldots, T-1$ **do**
3:     Recv_check_token_from_last_pipeline(GPU$_0$, GPU$_{N-1}$, $t_y$)
4:     **if** $t_y \neq t'_y$ **then**
5:         $i \leftarrow y + 1$
6:     **end if**
7:     prev_has_early_exited $\leftarrow$ False
8:     Run $M_0(x|x_{0,1,2,\ldots,i-1})$ in GPU$_0$
9:     Send_Activation_to_next_pipeline(GPU$_0$, GPU$_1$, $A_i^0$)
10:     wait_until_send_token()
11:     **if** reach_early_exit_point(GPU$_0$) **then**
12:         has_early_exited $\leftarrow$ True
13:         Sample $t'_{i+1}$ from $M_0(x|x_{0,1,2,\ldots,i-1})$
14:         Continue
15:     **end if**
16:     **if** not has_early_exited **then**
17:         Recv_token_from_other_pipeline(GPU$_0$, GPU$_x$, $t'_{i+1}$)
18:         wait_until_recv_token()
19:         Continue
20:     **end if**
21:     **for** $j \leftarrow 1, 2, 3, \ldots, N-1$ **do**
22:         Recv_Activation_from_prev_pipeline(GPU$_j$, GPU$_{j-1}$, $A_i^{j-1}$)
23:         wait_until_recv_token()
24:         Run $M_j(x|x_{0,1,2,\ldots,i-1})$ in GPU$_j$
25:         Send_Activation_to_next_pipeline(GPU$_j$, GPU$_{j+1}$, $A_i^j$)
26:         wait_until_send_token()
27:         **if** reach_early_exit_point(GPU$_j$) **and** not has_early_exited **then**
28:             has_early_exited $\leftarrow$ True
29:             Sample $t_{i+1}^j$ from $M_j(x|x_{0,1,2,\ldots,i-1})$
30:             Send_token_to_first_pipeline(GPU$_j$, GPU$_0$, $t_{i+1}^j$)
31:             wait_until_send_token()
32:         **end if**
33:         **if** $j = N-1$ **then**
34:             Sample $t_{i+1}$ from $M_{N-1}(x|x_{0,1,2,\ldots,i-1})$
35:             **if** not has_early_exited **then**
36:                 Send_token_to_first_pipeline(GPU$_{N-1}$, GPU$_0$, $t_{i+1}$)
37:                 wait_until_send_token()
38:             **else**
39:                 Send_check_token_to_first_pipeline(GPU$_{N-1}$, GPU$_0$, $t_{i+1}$)
40:             **end if**
41:         **end if**
42:     **end for**
43: **end for**

---

## E.2   EVALUATION DETAILS

We use OpenCompass to evaluate both the inference time and accuracy of PPSD under our proposed mechanism. For Vicuna models, the chat template is required for evaluation, while for LLaMA-2 models, we use the base template, as we do not adopt the chat version (i.e. LLaMA-2-Chat).

Experiments are conducted on three benchmarks: GSM8K (grade school math word problems) (Cobbe et al., 2021), HumanEval (code generation) Chen et al. (2021), and XSum (text summarization) Narayan et al. (2018). Following standard practice in prior speculative decoding works, we set the inference batch size to 1. The maximum number of generated tokens is fixed at 512 across all benchmarks.

Table 4: The hyperparameter values for early-exit heads training.For different models, we adopt different parallelism strategies to accelerate training process. DP refers to data parallelism. PP refers to pipeline parallelism.

| Model | Head | Learning Rate | Batch Size | Epoch | Seq Length | Parallelism |
|---|---|---|---|---|---|---|
| V 7B | Norm | 2e-5 | 16 | 1 | 2048 | DP=8 |
|  | Transformer | 5e-4 | 2 | 2 | 2048 | DP=8 |
| V 13B | Norm | 2e-5 | 16 | 1 | 2048 | DP=8 |
|  | Transformer | 5e-4 | 2 | 2 | 2048 | DP=8 |
| L 13B | Norm | 2e-5 | 16 | 1 | 2048 | DP=8 |
|  | Transformer | 5e-4 | 2 | 2 | 2048 | DP=8 |
| L 70B | Norm | 2e-5 | 16 | 1 | 4096 | DP=8 |
|  | Transformer | 5e-4 | 2 | 2 | 4096 | PP=8 |

Table 5: Model configurations.

| Model | # of Layers | Hidden Size | FFN Hidden Size |
|---|---|---|---|
| Vicuna-v1.5-7B | 32 | 4096 | 11008 |
| Vicuna-v1.5-13B | 40 | 5120 | 13824 |
| Llama-2-13B | 40 | 5120 | 13824 |
| Llama-2-70B | 80 | 8192 | 28672 |

# F  CASE STUDY

We further conduct a case study comparing the response pipeline of our proposed PPSD with the baseline EESD and auto-regressive decoding. As discussed in 3, speculative decoding with fixed draft lengths often incurs overhead due to invalid draft tokens. Our PPSD framework addresses this inefficiency through two key designs: pipeline-parallel early-exit execution and the verify-while-draft paradigm. As illustrated in 6 and 7, we present two qualitative examples from the XSum and Gsm8k benchmarks, respectively, to highlight the effectiveness and practical improvements achieved by our method.

# G  ADDITIONAL RESULTS

We present comprehensive results on the end-to-end inference time and speedup of Vicuna-7B and Vicuna-13B under various early-exit points, as summarized in 6 and 7, respectively. These experiments evaluate model performance across three benchmarks: XSUM, GSM8K, and HumanEval.

Table 6: Performance of Vicuna 7B exiting from different positions $E$. We use matrices such as acceptance ratio(AR), generation speed(GS), speedup(SR) to evaluate the effectiveness of PPSD.

| Model | $E$ | XSum | | | Gsm8k | | | HumanEval | | |
|---|---|---|---|---|---|---|---|---|---|---|
|  |  | AR | GS | SR | AR | GS | SR | AR | GS | SR |
| V 7B Norm | 32 | - | 29.32 | 1.00x | - | 25.07 | 1.00x | - | 31.36 | 1.04x |
|  | 24 | 82.03% | 32.95 | 1.12x | 85.64% | 29.08 | 1.16x | 87.45% | 36.34 | 1.21x |
|  | 16 | 38.48% | 34.71 | 1.18x | 54.54% | 33.83 | 1.35x | 64.46% | 44.71 | 1.49x |
|  | 8 | 32.34% | 38.24 | 1.30x | 32.26% | 32.71 | 1.30x | 49.11% | 43.45 | 1.44x |
| V 7B Trm | 32 | - | 29.32 | 1.00x | - | 25.07 | 1.00x | - | 30.66 | 1.00x |
|  | 24 | 86.90% | 31.03 | 1.06x | 90.90% | 27.60 | 1.10x | 92.37% | 31.63 | 1.03x |
|  | 16 | 78.05% | 42.66 | 1.45x | 83.29% | 38.31 | 1.53x | 88.27% | 43.24 | 1.41x |
|  | 8 | 67.38% | 53.53 | 1.83x | 71.64% | 48.80 | 1.95x | 80.56% | 70.32 | 2.29x |

[Input]
Document：Seven photographs taken in the Norfolk countryside by photographer Josh Olins will appear in the June edition.
……
Nicholas Cullinan, director of the National Portrait Gallery, said: \"Josh has captured the duchess exactly as she is - full of life, with a great sense of humour, thoughtful and intelligent, and in fact, very beautiful.\"
……
duchess is to visit the exhibition at the National Portrait Gallery on Wednesday, Kensington Palace said.\nBased on the previous text, provide a brief single summary:
The Duchess of Cambridge will feature on the cover of British Vogue to mark the magazine's centenary.

[AR outputs]
The Duchess of Cambridge has been photographed for the June edition of British Vogue.

[EESD outputs]
ROUND0: The ~~photographer will appear~~ Duchess
ROUND1: of Cambridge has ~~to~~ been
ROUND2: ~~taken~~ photographed
ROUND3: for the ~~Vogue 100~~ June
ROUND4: edition ~~\n~~ of
ROUND5: British Vogue.

[PPSD outputs]
The ~~photographer~~ Duchess of Cambridge has ~~to~~ been ~~token~~ photographed for the June edition of British Vogue.

Figure 6: A visualization of the generation process from XSum dataset, including the input text, the text generated by the original LLM using auto-regression strategy, the generation process of EESD, and the verify-while-draft process by PPSD. In SD generation process, the green color represents the draft token that are accept by LLM, and the red color represents the rejected draft tokens, and blue color represents the tokens that will be sampled from full model forward in verification.

Table 7: Performance of Vicuna 13B exiting from different positions $E$.

| Model | $E$ | XSum | | | Gsm8k | | | HumanEval | | |
|---|---|---|---|---|---|---|---|---|---|---|
| | | AR | GS | SR | AR | GS | SR | AR | GS | SR |
| V 13B Norm | 40 | - | 19.32 | 1.00x | - | 16.31 | 1.00x | - | 19.24 | 1.00x |
| | 30 | 84.00% | 20.90 | 1.08x | 85.91% | 18.71 | 1.15x | 89.30% | 22.90 | 1.19x |
| | 20 | 62.70% | 26.53 | 1.37x | 68.44% | 24.10 | 1.48x | 74.54% | 30.10 | 1.56x |
| | 10 | 36.67% | 22.51 | 1.17x | 42.15% | 20.24 | 1.24x | 51.81% | 25.80 | 1.34x |
| V 13B Trm | 40 | - | 29.62 | 1.00x | - | 25.83 | 1.00x | - | 28.98 | 1.00x |
| | 30 | 88.00% | 34.16 | 1.15x | 91.58% | 30.67 | 1.19x | 93.93% | 34.13 | 1.18x |
| | 20 | 76.00% | 47.08 | 1.59x | 85.23% | 43.14 | 1.67x | 90.64% | 54.21 | 1.87x |
| | 10 | 68.32% | 57.77 | 1.95x | 70.87% | 53.38 | 2.07x | 81.28% | 74.69 | 2.58x |

For the Vicuna-7B Norm model, we observe that the highest speedup is consistently achieved when the model exits at half the number of total layers (e.g., $E = 16$), with an average speedup ratio of up to $1.35\times$ across tasks, while still maintaining reasonable acceptance and generation quality. For instance, on the GSM8K task, exiting at $E = 16$ yields a 54.54% acceptance rate and a $\mathbf{1}.35\times$ speedup, with only a moderate drop in generation score compared to the full model. In contrast, the Vicuna-7B Transformerhead model exhibits a different behavior: the optimal early-exit point appears earlier, typically around one-quarter of the total layers (e.g., $E = 8$). In this case, the model achieves significantly higher speedups — up to $2.29\times$ on HumanEval — while maintaining

[Input]
Question: A candle melts by 2 centimeters every hour that it burns. How many centimeters shorter will a candle be after burning from 1:00 PM to 5:00 PM?\nLet's think step by step\nAnswer:

[AR outputs]
A candle melts by 2 centimeters every hour that it burns.\nThe candle burns for 4 hours, so it will melt by 2*4=8 centimeters.\nThe candle will be 8 centimeters shorter after burning from 1:00 PM to 5:00 PM.\nThe answer is 8.

[EESD outputs]
ROUND0: ~~For the First three~~ A
ROUND1: ~~answer is 14 .~~ candle
ROUND2: ~~will be 2 minutes~~ melts
ROUND3: by 2 ~~minutes.~~ centimeters
ROUND4: ~~shorter after burning~~ every
ROUND5: hour that it burns .
ROUND6: \nThe candle ~~melts~~ burns
ROUND7: for 4 hours ~~and~~ ,
ROUND8: ~~The~~ so
ROUND9: it will melt ~~down~~ by
ROUND10: 2*4=~~10~~ 8
ROUND11: centimeters.\nThe candle
ROUND12: will be 8 centimeters shorter
ROUND13: ~~than~~ after
ROUND14: burning from 1:00
ROUND15: PM to 5:00
ROUND16: PM.\n The answer
ROUND17: is ~~20~~ 8
ROUND18: .

[PPSD outputs]
~~The~~ A ~~answer~~ candle ~~will~~ melts by 2 ~~minutes~~ centimeters ~~shorter~~ every hour that it burns.\nThe candle ~~melts~~ burns for 4 hours ~~and~~ , ~~The~~ so it will melt ~~down~~ by 2*4=~~10~~ 8 centimeters.\nThe candle will be 8 centimeters shorter ~~than~~ after burning from 1:00 PM to 5:00 PM.\nThe answer is ~~20~~ 8.

Figure 7: A visualization of the generation process from Gsm8k dataset, including the input text, the text generated by the original LLM using auto-regression strategy, the generation process of EESD, and the verify-while-draft process by PPSD.

strong acceptance rates (e.g., 80.56%) and competitive generation scores. This suggests that Transformerhead is better suited for aggressive early-exiting without substantial loss in performance. A similar trend is observed for the Vicuna-13B models. For the Norm variant, a balanced performance is achieved at $E = 20$, with speedup ratios reaching up to $1.48\times$ on GSM8K and $1.56\times$ on HumanEval, along with acceptance rates above 60%. For the Transformerhead variant, earlier exits (e.g., $E = 10$) again lead to the highest speedups — up to $2.58\times$ on HumanEval — while still achieving an 81.28% acceptance rate and strong output quality. Notably, the overall average speedup (OA) across all tasks reaches $1.87\times$ at this exit point.

These results highlight the architecture-dependent nature of optimal exit points. While standard models benefit most from mid-layer exits, Transformerhead variants allow for much earlier exits with minimal degradation. This trade-off can be quantitatively understood using the throughput former 6, which compares parallel decoding efficiency (PPSD) against traditional auto-regressive decoding. Therefore, maximizing the overall speedup requires balancing the acceptance rate and the layer depth of exits. The Transformerhead model's ability to achieve high acceptance rates even at shallow layers leads to significantly higher $\rho^\star$, offering a favorable trade-off between efficiency and accuracy. These findings offer actionable insights into deploying early-exit strategies that optimize for both model performance and system-level throughput.

