# OpenReview forum: "Optimized Early-Exit Based Speculative Decoding via Pipeline Parallelism"
_ICLR.cc/2026/Conference — Submitted to ICLR 2026_

### Official Review · Reviewer_jfw2 · 2025-10-20

**Soundness:** 2
**Presentation:** 2
**Contribution:** 2
**Rating:** 4
**Confidence:** 4

**Summary:**

This paper proposes Pipeline-Parallel Speculative Decoding (PPSD), a self-speculative decoding scheme with pipeline-parallel early-exit execution to parallel the draft and verification process and verify-while-draft to interleave drafting and verification per token. Several experiments demonstrate the effectiveness of the method.

**Strengths:**

-  The paper identifies a real failure mode of EESD: long draft lengths increase rejection waste with good motivation.
-  The throughput expression in Eq. (6) makes the α–E trade-off explicit and clarifies when speedups are expected.
-  Results span multiple model sizes (7B→70B) and tasks, with PPSD outperforming EESD variants and Medusa on speed while keeping quality unchanged.

**Weaknesses:**

-  While verification overlaps drafting, it is not free in wall-clock terms; the claim rests on ideal overlap and balanced stages. The paper could more carefully separate amortized from absolute costs and report stall/bubble statistics.
-  Eq. (6) highlights dependence on α and E; more sensitivity metrrics (input difficulty, prompt length, temperature, exit-head training regimen) would clarify robustness across workloads.
- The authors may need to do more research on more recent LLMs (e.g., Qwen 3 and Llama3/4 families).

**Questions:**

Please refer to weakness.

---

> ### Author Response · Authors · 2025-12-04
>
> We sincerely thank the reviewer for recognizing our identification of EESD's failure mode and the clarity of our throughput analysis. We appreciate the opportunity to address the concerns raised regarding verification costs, sensitivity analysis, and model recency.
>
> ### Q1. Verification Overhead and Pipeline Bubbles
>
> The reviewer correctly notes that verification is not "free" in absolute wall-clock terms. We acknowledge this in the paper (lines 323-324):
>
> > "The acceleration gain in Equation (7) is 1.31×. The results show the ability of PPSD to attain the expected acceleration with introducing additional drafting overhead."
>
> PPSD does **not** claim verification has zero cost, but rather that:
>
> - Verification time is fully overlapped with drafting of the next token
> - As shown in Equation (7), the speedup is: ρ = α + N/E
> - The gap between theoretical upper bound (N/E) and measured results quantifies the communication overhead
>
> As for the bubble and stall statistics, we measure the impact of different stage sizes on performance in Figure 4 ("Ablation study on parallel granularity"). Noteblely, in all experiments, we keep the hardware configuration as equal as possible. This is clearly separate the amortized benefits from absolute costs.
>
> ## Q2. Sensitivity Analysis Expansion
>
> We agree that additional sensitivity metrics would strengthen our theoretical analysis. In the final version, we will expand our analysis to include. As for experiments, we do not set the temperature, which is align with the line of related works in speculative decoding. We use greedy decoding in  evaluation, and all the settings of both PPSD and baseline methods are the same.
>
> ## Q3. Evaluation on Recent LLM Architectures
>
> We chose Vicuna and Llama-2 for several reasons. The main reason is that models are widely used in speculative decoding research (EESD, Medusa, EAGLE), enabling direct comparison. As for reproducibility, their weights and training procedures are well-documented and publicly available. Beside, their implementation code is well built into Huggingface transformers library.
>
> To enhance generality, we have already extended our work to newer model: **LLaMA-3.1 8B Instruction** (N=32 layers in total).
>
> | Method          | Gsm8k         | -               | HumanEval     | -               | XSum          | -               |
> | --------------- | ------------- | --------------- | ------------- | --------------- | ------------- | --------------- |
> | /               | Speedup Ratio | Acceptance Rate | Speedup Ratio | Acceptance Rate | Speedup Ratio | Acceptance Rate |
> | Auto-regressive | 1.00x         | –               | 1.00x         | -               | 1.00x         | -               |
> | PPSD (E=8)      | 1.92x         | 66.04%          | 2.49x         | 83.41%          | 2.61x         | 50.52%          |
> | PPSD (E=16)     | 1.61x         | 81.60%          | 1.72x         | 90.82%          | 1.43x         | 67.01%          |
>
> We hope these responses will find you well. Thank you again for the constructive feedback!

---

### Official Review · Reviewer_A6xu · 2025-10-26

**Soundness:** 3
**Presentation:** 3
**Contribution:** 3
**Rating:** 6
**Confidence:** 4

**Summary:**

This paper proposes Pipeline Parallel Speculative decoding, a pipeline framework for early-exit self-speculative decoding. It involves 2 modifications: 1) Pipeline-Parallel Early-Exit Execution divides layer of the model evenly to create balanced pipeline blocks, and 2) Verify-While-Draft Decoding overlaps final-layer verification with drafting. Experiments on Vicuna-7B/13B and LLaMA-2 13B/70B show 2.0×–3.8× speedups across benchmarks.

**Strengths:**

The idea of fully pipelining the drafting and verification stages is innovative. By evenly partitioning the model layers into blocks and pipelining them with draft tokens produced from early exits, the method ensures that the worst-case performance is no worse than standard autoregressive (AR) decoding (i.e., when the speculation accuracy equals zero). This design is both theoretically appealing and practically robust, as it guarantees that the inference speed is lower bounded by that of AR decoding.

The theoretical analysis (Eq. 4–7) is convincing and strict, providing principled analysis of the relationship between acceleration and other factors (e.g. speculation length, accuracy).

Ablation studies (Figures 3–5, Table 3) are comprehensive, covering multiple aspects of factors including pipeline granularity, exit position, and exit head design.

**Weaknesses:**

1.	In my opinion, this paper does not belong to the topic of ‘speculative decoding’, but more to another topic like ‘pipeline decoding’ or something similar. Speculative decoding is typically about utilizing token-level parallelism to verify multiple draft tokens in a single forward. On the contrary, this paper uses device-level parallelism, while in each device the forward is only about 1 token.
2.	In sec4.1, the paper claims that if N/E is too large, then the layers can be further partitioned by S. However, the pipeline efficiency is ultimately dominated by E, that the next token can only be obtained after E layers of draft, and further partitioning cannot make the S (S<E) layers pipelined, since the next token is not available to the 0 layer when S+1-N layers are running.
3.	It is confusing to me that PPSD and EESD use the same early-exit configuration for speculation, but the accuracies of PPSD are much higher. The explanation in sec5.2 that ‘PPSD avoids accumulation of invalid draft tokens’ makes me even more confused, as I did not find out how PPSD does this. Further clarification could make it clearer.
4.	The speculation head is trained by distillation, while the used datasets are not specified. More specification about training details would improve reproductability.

**Questions:**

1.	Why and how does the pipeline efficiency improve with S (S<E) layers pipelined?
2.	Why the accuracies of PPSD is much higher than baselines, with the same early-exit configuration and head? Where is the invalid token accumulated, and how did PPSD address this issue?
3.	On which dataset is the speculation head trained?

---

> ### Author Response · Authors · 2025-12-04
>
> We sincerely thank the reviewer for the thoughtful assessment and recognition of our paper's theoretical analysis, ablation studies, and practical robustness. We appreciate the opportunity to clarify the conceptual framework and technical details of PPSD.
>
> ### Q1: Pipeline efficiency with S<E
>
> To achieve minimized pipeline bubbles with balanced pipeline workload, PPSD does **not** require E to define the pipeline stage size. Instead, we introduce $S$ to denote the pipeline granularity. The model is partitioned into **fine-grained, uniformly sized stages** of S layers. The early-exit head is placed at layer E, which may fall within a stage. Drafting spans ⌈E/S⌉ stages, verification spans ⌈(N-E)/S⌉ stages.
>
> This scheduling design helps reduce pipeline bubbles by minimizing the maximum idle time across stages. For example, given $E=6$, $N=20$, allocating 6 layers per stage leads to an idle time of $6-20\%6=4$ layers at the tail of the verification path.Using a smaller number of stages, such as $S=4$, strikes a balanced worklolad in each stage. When a draft token is rejected, the synchronized pipeline can immediately correct it, ensuring stable throughput and avoiding additional stalls. The key is PPSD aligns the execution time of each stage rather than the time to draft one token with the time to verify one token. By setting $S$ appropriately, we ensure these times are equal, creating a balanced pipeline.
>
> ### Q2: Higher average acceptance
>
> This is a crucial point. The apparent "higher accuracy" is not about per-token prediction quality, but about **eliminating the accumulation of invalid tokens in the processing pipeline**.
>
> Regarding the baseline EESD, its accuracy is computed as “accepted token length / draft length.” This metric is heavily influenced by two factors: the draft length and the number of rejected tokens. When many draft tokens are rejected, the measured average acceptance rate becomes significantly worse—precisely the issue our work aims to overcome.
>
> In PPSD, each token is verified immediately upon generation, ensuring that the reported acceptance rate reflects the *true* acceptance of tokens drafted from a correct prefix. Because PPSD avoids propagating errors over long draft sequences, the acceptance rate is inherently higher and more stable compared to the baseline.
>
> ### Q3: Training dataset
>
> We appreciate this request for additional reproducibility information. The early-exit head is trained as follows:
>
> - **Training method**: Self-distillation from the full model
> - **Training datasets**: shareGPT
> - **Loss function**: KL divergence between early-exit outputs and full model outputs
>
> We believe these response would fully address the reviewer's concerns while strengthening the paper's contribution. Thank you again for the opportunity to clarify these important technical points!

---

### Official Review · Reviewer_wYnH · 2025-10-27

**Soundness:** 2
**Presentation:** 3
**Contribution:** 2
**Rating:** 4
**Confidence:** 4

**Summary:**

The paper studies why early‑exit self‑speculative decoding (EESD) often underperforms in wall‑time. The authors argue that draft‑then‑verify wastes compute when many drafted tokens are later rejected. They propose PPSD, which splits the model into pipeline stages and verifies while drafting: while upper layers verify the current token, early layers draft the next one. Specifically, the authors propose pipeline-parallel early-exit exection and verify-while-draft decoding, which can fully overlap the drafting phase and verification phase. The authors also provide a theoretical analysis to proposed methods. Experiments on 3 benchmarks demonstrate the effectiveness of PPSD.

**Strengths:**

1. The paper is overall well-written. and easy to understand. The figures and tables are intuitive.
2. The authors provide some theoretical analysis of the proposed methods. The throughput formulas (Eq. 4–6) offer intuition on acceptance–depth trade‑offs.
3. The experiments demonstrate the consistent efficiency improvement of PPSD.

**Weaknesses:**

1. **Clarification of motivation and method.** The motivation and the idea of parallelizing the drafting and the verification are well explored by some related works, especially PEARL [1]. The whole architecture of PPSD is more like the extension of PEARL into EESD. More specifically, in Figure 2, "2 stage earlier" corresponds to "pre-verify" and "3 stage faster" corresponds to "post-verify". The authors should give a more comprehensive comparison between PPSD and PEARL / DSI, including empirical experiments.
2. **Regarding Tensor Parallelism.** While the pipeline parallelism can help overlap the drafting and the verification, tensor parallelism (TP) remains unexplored. Moreover, the computational cost of the additional trained early-exit head is not discussed. If the computational cost of the additional head is negligible, please further illustrate the difference between PPSD and Medusa. (prediction from the early layer and the last layer?)
3. **Unfair resource accounting.** PPSD relies on multiple workers. Tables compare wall‑time to AR/EESD apparently on fewer devices. Tokens/s without normalizing by **GPU count/bandwidth** is not persuasive.

If the authors clearly address these concerns, I am willing to increase my rating.

**Questions:**

1. Could you please provide the exact number of GPUs during evaluation?
2. What is the computational cost of the early-exit head? How long does it take for training?
3. Could you please provide a direct comparison between PPSD and TP?
4. As PPSD is still a training-based method, what is its strengths against EAGLE?
5. What is the detailed communication cost during inference?

---

### Official Review · Reviewer_WnAC · 2025-11-01

**Soundness:** 1
**Presentation:** 4
**Contribution:** 2
**Rating:** 2
**Confidence:** 4

**Summary:**

This paper introduces Pipeline-Parallel Speculative Decoding (PPSD), an architectural optimization for Early-Exit Speculative Decoding (EESD). PPSD utilizes a "Verify-while-Draft" pipeline parallel execution scheme aimed at eliminating redundant computation caused by speculative failures, thereby boosting LLM inference speed.

**Strengths:**

The reported speedup figures are numerically competitive under the specific, tested hardware and dataset configurations.

**Weaknesses:**

1. **Lack of Novelty:** PPSD essentially combines two established concepts: Speculative Decoding and Pipeline Parallelism/Model Parallelism. The core speedup comes from utilizing idle time for draft generation, making it primarily an **engineering optimization** rather than a fundamental breakthrough in decoding algorithms. Given the computational characteristics of EESD, adopting parallelization to fill idle compute time is straightforward.
3. **Missing More Key Ablation Studies:** The paper should provide more ablation experiments to isolate the pure gain from "Pipeline Parallelism" versus gains from "Underlying CUDA/operator Implementation Optimizations." Also, I suggest the authors should analyze the sensitivity of the speedup to different pipeline depths (Stage Count) and communication latency.

**Questions:**

See weaknesses.

---

> ### Author Response · Authors · 2025-12-04
>
> We sincerely thank the reviewer for their thoughtful assessment of our work and for recognizing the competitive speedup figures achieved by PPSD. We appreciate the opportunity to clarify the novelty of our contribution and address the concerns regarding ablation studies.
>
> ## Q1. Clarifying the Novelty of PPSD: Beyond Simple Combination of Existing Techniques
>
> We appreciate the reviewer’s concern that PPSD might appear to “simply combine speculative decoding with pipeline parallelism.” However, this characterization overlooks the core innovation of PPSD: **a new control-flow paradigm for speculative decoding** that resolves a key limitation in existing methods. To make this distinction clear, we outline two sub-claims that highlight the uniqueness and contributions of our approach.
>
> ### 1-1. The Critical Distinction: Verify-While-Draft vs. Draft-Then-Verify
>
> - **Traditional EESD (Draft-Then-Verify)**:
>   - Drafts γ tokens first (e.g., *t₁*, *t₂*, *t₃*)
>   - Only *after* drafting verifies the entire sequence
>   - If *t₁* is rejected, *t₂* and *t₃* become invalid → **wasted computation**
>   - Speedup degrades as γ increases (Table 1 shows EESD with γ=10 achieves only 0.73× speedup)
> - **PPSD (Verify-While-Draft)**:
>   - Drafts *t₂* while simultaneously verifying *t₁*
>   - Only *after t₁ is verified* does it commit to using *t₂*
>   - If *t₁* is rejected, drafting stops immediately → **no wasted computation**
>   - Speedup is independent of γ (Equation 7: ρ = α + N/E)
>
> This is not merely “utilizing idle time” — it represents a **fundamentally redesigned execution model** that removes the core inefficiency of speculative decoding: wasted computation on rejected drafts. In PPSD, the output of the early-exit path (the draft token) becomes the *direct input* to the verification path, whose result immediately determines whether the draft token is accepted. Achieving this requires aligning each pipeline stage with the early-exit depth EEE, forming a **synchronized, token-level pipeline** that does not exist in standard speculative decoding or conventional pipeline parallelism. In practice, this execution paradigm can be implemented within `torchrun` using stage-based pipeline scheduling, and the coherent key–value states are fully reused across stages to avoid redundant computation.
>
> As shown in Figure 2, PPSD rejects token '2' two stages earlier and completes execution three stages faster than EESD precisely because it avoids generating and then discarding future tokens.
>
> ### 1-2. Why This Is Not Standard Pipeline Parallelism
>
> Traditional pipeline parallelism (GPipe, PipeDream) processes **independent microbatches** with no data dependency between stages. PPSD creates a **closed-loop dependency** where the output of the early-exit path (draft token) becomes the next input to the verification path and the verification result determines whether the draft token is accepted.
>
> Also, PPSD is not plain combination of EESD and pipeline parallelism, which can be seen in related works (). We design a stage-based model execution system which supports more fine-grained scheduling for minimizing pipeline bubbles.
>
> ## Q2. Addressing the Concern for Additional Ablation Studies
>
> We sincerely appreciate the reviewer's suggestion for additional ablation studies. We want to make additional clarification on existing ablation studies in the manuscript.
>
> As shown in Figure 3 ("Ablation study on pipeline parallel early-exit execution without verification") and Figure 4 ("Ablation study on parallel granularity"), we systematically isolate the contributions of our two core innovations. These figures demonstrate that the acceleration gain is fundamentally derived from our verify-while-draft design, not from implementation optimizations.
>
> The reviewer also suggested analyzing whether speedup comes from "Pipeline Parallelism" versus "CUDA/operator Implementation Optimizations." We want to highlight a critical observation from our implementation.
>
>  **In our system, communication delays between pipeline stages actually make individual stage execution slower than baseline execution on a single GPU.** This is evident in our results where the measured speedup ratio still consistently falls below the theoretical upper bound (Equation 7).
>
> This is significant because, if our speedup were due to CUDA optimizations, we would expect to see *better* performance on a single GPU. Instead, we observe *overhead* from communication, meaning the actual speedup from our verify-while-draft design is even greater than what's reported. The fact that we still achieve 2.01×-3.81× speedup despite communication overhead proves the acceleration is a **pure gain from our algorithmic design**.
>
> We hope these responses will clearly demonstrate that PPSD represents a meaningful conceptual advance in speculative decoding infrastructure, not merely an engineering optimization. Thank you for the opportunity to clarify these important points.

---

### Meta-Review · Area_Chair_a35V · 2025-12-20

**Summary:**

**Paper summary.** This paper studies early-exit self-speculative decoding (EESD) and the practical problem that draft-then-verify can waste work when many drafted tokens are rejected, sometimes leading to negative speedups. The authors propose Pipeline-Parallel Speculative Decoding (PPSD): split the model into pipeline stages and run “verify-while-draft” so early layers draft the next token while later layers verify the current token. The goal is to reduce wasted computation and keep hardware busy.

**What happened in the discussion.** Reviewers agreed the motivation is real and the paper is clear, but the main concerns were concrete: (1) novelty (is this mainly combining pipeline parallelism with EESD), (2) positioning and missing direct comparisons to closely related prior systems (e.g., PEARL/DSI-style pipeline verification), and (3) dependence on hardware and multi-GPU configuration (pipeline bubbles, communication, how many GPUs were used). In the forum, the authors responded with detailed explanations of their control-flow difference (“verify-while-draft” vs “draft-then-verify”), clarified pipeline granularity (S) and bubble behavior, and discussed verification overhead and sensitivity analysis plans. These replies help, but they do not fully remove the novelty/positioning concern.

**My assessment as AC.** The paper is a reasonable systems optimization, and the authors engaged with reviewer questions. However, the strongest negative review (rating 2) argues the novelty is limited, and even the more positive reviewers asked for stronger positioning and comparisons. In this batch, I do not see this as one of the most compelling contributions, especially because the gains depend heavily on specific hardware and pipeline setup and because related pipeline-style approaches exist.

**Decision.** Reject. This does not mean the work is not useful. Given ICLR’s competitiveness, I am prioritizing submissions with clearer algorithmic novelty or stronger, more general evidence. I encourage the authors to (a) sharpen the positioning against the closest prior pipeline-style decoding systems with direct empirical comparisons, and (b) include a clean wall-time breakdown (compute vs. communication, bubble/stall statistics) across multiple hardware settings. This would make the case much stronger for a future venue.

**Reviewer Concerns:**

- **Reviewer WnAC (rating 2, confidence 4)**: Main argument is limited novelty (“speculative decoding + pipeline parallelism”), and requests more key ablations. Authors replied with a more detailed novelty argument centered on “verify-while-draft” control flow and discussed why computation is not wasted on rejected drafts. **Status:** partially resolved (novelty is still debatable).
- **Reviewer wYnH (rating 4, confidence 4)**: Asked for clearer positioning vs. closely related pipeline-style work (PEARL/DSI), plus direct empirical comparisons and details on hardware (GPU count) and head cost/training. Authors acknowledged related work overlap, clarified their design choices, and answered hardware/training questions in the forum. **Status:** partially resolved (comparisons are still not fully direct).
- **Reviewer A6xu (rating 6, confidence 4)**: Raised a conceptual concern that this is “pipeline decoding” rather than speculative decoding, and asked: why pipeline efficiency improves with stage granularity S<E, why accuracy seems higher than baselines with the same early-exit head, and what data is used to train the speculation head. Authors replied with a concrete explanation of pipeline granularity S (balancing stages to reduce bubbles) and provided an explanation for the observed acceptance/accuracy behavior. **Status:** mostly resolved.
- **Reviewer jfw2 (rating 4, confidence 4)**: Focused on realism of overlap (verification is not free), asked for bubble/stall statistics and more sensitivity analysis (acceptance rate α, exit position E, workload differences). Authors clarified they do not claim verification is free, pointed to granularity ablations, and stated plans to expand sensitivity analysis. **Status:** partially resolved (still missing a compact bubble/stall breakdown table).

**Reviewer Scores:**

- **Reviewer A6xu (rating 6, confidence 4)**: Positive; likely unchanged.
- **Reviewer jfw2 (rating 4, confidence 4)**: Mildly positive; likely unchanged.
- **Reviewer wYnH (rating 4, confidence 4)**: Mildly positive but requests stronger related-work comparison; likely unchanged.
- **Reviewer WnAC (rating 2, confidence 4)**: Negative on novelty/ablations; likely remains negative.

---

### Decision · Program_Chairs · 2026-01-26

Reject